# Detail Enhancement and Transfer Balance for Open-Vocabulary Compositional Zero-Shot Learning

## Abstract

Compositional Zero-Shot Learning (CZSL) aims to recognize unseen attribute-object compositions by learning from seen combinations of visual primitives. Recent advances extend this task to the Open-Vocabulary setting (OV-CZSL), where novel attributes or objects may appear at test time. This setting presents two major challenges: (1) global visual features often lack the granularity required to distinguish fine-grained attribute information, particularly in unseen compositions; and (2) indiscriminate knowledge transfer from seen to unseen compositions can compromise class boundaries, leading to overfitting on seen compositions. To address these issues, we propose a novel OV-CZSL framework that integrates Detail Enhancement and Transfer Balance (DETB). Specifically, we propose a Multi-scale Condition-guided Diffusion (MCD) module that selectively refines challenging samples by integrating global semantic priors with localized visual disentangled representations, enabling the recovery of fine-grained attribute information essential for compositional recognition. Furthermore, we introduce a Transfer Balance Loss (TBL) that adaptively adjusts the semantic margins between seen and unseen compositions according to their inter-class similarity. This encourages effective knowledge transfer while maintaining clear class separation. Extensive experiments on three OV-CZSL benchmark datasets show that DETB consistently outperforms existing approaches, setting a new state-of-the-art.

## 1 Introduction

Humans naturally possess the ability to generalize learned concepts to construct novel compositions. For example, given prior knowledge of a 'yellow bird' and a 'red flower', humans can effortlessly infer the meaning of an unknown composition such as a 'yellow flower'. Inspired by it, **C**ompositional **Z**ero-**S**hot **L**earning (CZSL) aims to classify images into unseen attribute-object pair labels by learning primitives (*i.e.*, attributes or objects) from images with known pair labels. However, CZSL is constrained by the limited set of primitives observed during training, which restricts its ability to generalize to unseen attributes, objects, and their compositions. Based upon this setting, recently a more challenging task called **O**pen **V**ocabulary-**C**ompositional **Z**ero-**S**hot **L**earning (OV-CZSL) extends compositional reasoning by introducing novel primitives, aligning more closely with real-world recognition scenarios.

Recent advances in CZSL highlight the importance of disentangling visual representations of attributes and objects, as their co-occurrence in training images leads to entangled features that limit compositional generalization. A widely adopted solution is the three-branch framework Saini et al. (2022); Hao et al. (2023); Wang et al. (2023b); Huang et al. (2024), which employs attribute and object branches to learn separate visual representations aligned with their respective textual embeddings, while a composition branch performs final prediction based on global visual-textual similarity. By extending this paradigm to the Open-Vocabulary setting (OV-CZSL), the most recent method BSPC Saini et al. (2024) inherits the strengths of this framework and further incorporates external knowledge (*e.g.*, word embeddings) to bridge the semantic gap between seen and unseen attribute-object pairs, thereby enabling the transfer of primitive knowledge. However, BSPC still suffers from two key limitations in the OV-CZSL setting.

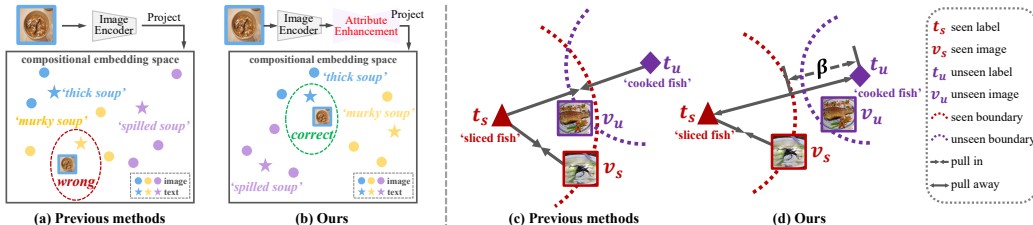

Figure 1: (a) As fine-grained semantic concepts, attributes are prone to misclassification (*e.g.*, 'thick' misidentified as 'murky') when models solely rely on global features with insufficient local detail. (b) We mitigate this issue by employing diffusion-based enhancement to recover attribute-specific details, thereby preserving fine-grained visual granularity. (c) Previous methods facilitate knowledge transfer via similarity-based consistency, which blurs seen-unseen decision boundaries, leading to misclassification of unseen samples $v_u$ as the seen composition 'sliced fish' $t_s$. (d) We alleviate it by introducing a class-adaptive balancing distance $\beta$, enabling correct prediction as 'cooked fish' $t_u$.

**(1) Lack of attribute-relevant visual details in global representations.** Intuitively, attributes often reflect fine-grained semantics (*e.g.*, color, material, or state), which are usually difficult to capture by global visual features extracted by standard backbones Huynh & Elhamifar (2020). OADis Saini et al. (2022) attempts to address this by leveraging samples with the same attribute to enhance feature disentanglement in the attribute branch. However, most CZSL methods, including OADis, still fail to preserve these fine-grained details in the composition branch, resulting in misclassification during final prediction. In OV-CZSL, this issue becomes more pronounced, as unseen attributes appear at test time are more easily confused with seen ones due to limited visual granularity, as illustrated in Figure 1 (a). To address this issue, ***enhancing the attribute-relevant details absent from global visual features*** is essential to further improve the model's ability to distinguish both seen and unseen compositions.

**(2) Indiscriminate knowledge transfer compromises class boundaries.** To enhance knowledge transfer from seen to unseen compositions for better generality, BSPC aligns seen and unseen compositions by measuring the similarity of their compositional word embeddings. However, this paradigm may compromise class boundaries and cause the model to overfit to seen compositions, as shown in Figure 1 (c). Therefore, another challenge in OV-CZSL lies in ***balancing knowledge transfer and discrimination***: how to model the relationships between seen samples, seen labels, and unseen labels in a shared space, while ensuring knowledge transfer is maintained without sacrificing discriminative ability. In particular, preventing undesired alignment between unseen visual features and seen textual features is essential for robust generalization to novel compositions.

To address the two core limitations discussed above, we propose a novel OV-CZSL framework named DETB, which aims to achieve **D**etail **E**nhancement and **T**ransfer **B**alance. Recent studies Li et al. (2023); Wu et al. (2024) have shown that diffusion models Ho et al. (2020) possess strong capabilities in refining visual details (*e.g.*, textures and edges) by iterative denoising, which have superior performance in tasks like super-resolution, image deblurring, and image inpainting. Motivated by these findings, we design a Multi-scale Condition-guided Diffusion (MCD) module to enhance attribute representations. Specifically, we first identify a subset of hard-to-predict samples by filtering those with low attribute prediction confidence, which are then fed into MCD to enhance their attribute representations. Given the strong contextual dependence of attribute features, MCD integrates both global semantic context and local disentangled visual features to generate refined and discriminative attribute representations.

To balance knowledge transfer and category discrimination, we introduce a class-adaptive margin between each seen composition and its top-$k$ most similar unseen compositions, as shown in Figure 1 (d). We define the margin as a function of semantic similarity, allowing closer pairs to maintain smaller margins. We further propose a transfer balance loss (TBL), which explicitly constrains the class-aware margins, encouraging the model to establish clear decision boundaries while supporting effective semantic transfer.

Our main contributions are summarized below:

- We propose DETB, a novel OV-CZSL framework to achieve detail enhancement and transfer balance. To our knowledge, we are the first to leverage the strong generative ability of diffusion models for fine-grained detail recovery in both CZSL and OV-CZSL.

- We propose a multi-scale condition-guided diffusion (MCD) module to enhance attribute representations for hard samples, guided by both global context and local disentangled features. Furthermore, a class-adaptive transfer balance loss (TBL) dynamically adjusts margins based on semantic similarity, promoting clearer class boundaries between confusing seen and unseen compositions.

- We comprehensively evaluate our DETB on three OV-CZSL datasets, achieving state-of-the-art performance.

## 2 RELATED WORKS

**Compositional Zero-Shot Learning (CZSL)** represents a specialized branch of Zero-Shot Learning (ZSL). It aims to classify images into previously novel attribute-object pair labels by learning primitives (*i.e.*, attributes or objects) in images with known pair labels. Early studies on CZSL mainly follow two paradigms: word composition and visual disentanglement. Some word composition methods Naeem et al. (2021); Mancini et al. (2022); Karthik et al. (2022) learn joint representations via graph convolutional networks, leveraging the dependencies among attributes, objects, and their compositions to facilitate knowledge transfer from seen to unseen pairs. Recent works Nayak et al. (2022); Lu et al. (2023) utilize CLIP Radford et al. (2021) to learn soft prompts for individual primitives, which are then combined into novel compositional prompts. For visual disentanglement, contrastive-based approaches Wei et al. (2019); Yang et al. (2020) design attribute-object contrastive losses to improve representation separability. Other methods Saini et al. (2022); Hao et al. (2023) adopt attention-based mechanisms to retrieve samples sharing primitives, enabling the extraction of disentangled features for better separation.

However, CZSL methods always depend on a fixed set of seen attributes and objects. To better model open-world conditions, Open-Vocabulary CZSL (OV-CZSL) has emerged Saini et al. (2024), allowing unseen attributes and objects at test time. It leads to challenges like semantic drift and stronger entanglement, as new primitives lack visual supervision. Our work follows this direction by tackling OV-CZSL's core issues: enhancing attribute discriminative details and mitigating overfitting to seen classes.

**Feature Disentanglement** focuses on separating latent semantic factors (*e.g.*, style, identity, attributes) to improve generalization across various visual tasks. In domain generalization methods Zhang et al. (2022); Nguyen et al. (2021), they isolate domain-invariant features to reduce distribution shift. In face recognition methods Tran et al. (2017); Zhang et al. (2021), they disentangle identity from confounding factors like pose and age. In few-shot learning methods Xu et al. (2021); Cheng et al. (2024), they separate class-generic and specific features to improve transfer with limited data. These works demonstrate that learning factorized representations is broadly beneficial for robust and compositional visual understanding.

In CZSL, visual feature disentanglement of attributes and objects has become a prevailing strategy. Our work aims to advance this direction by extracting attribute-disentangled features with improved accuracy and richer details.

**Diffusion Models** Ho et al. (2020), originally designed for image generation, have recently been adapted for feature reinforcement in discriminative tasks by leveraging their ability to iteratively denoise and generate structured representations. In semantic segmentation, Zbinden et al. (2023) enhances feature learning by generating diverse segmentation masks conditioned on images. Diffu-Mask Wu et al. (2023) leverages Stable Diffusion's cross-attention to synthesize pixel-level annotated images for supervision. DFormer Wang et al. (2023a) injects noise into ground-truth masks and denoises them to improve universal segmentation.

These works demonstrate how diffusion models can enrich feature representations across tasks through diverse, semantically guided synthesis. In this paper, we leverage the powerful detail-capturing capacity of diffusion models and propose a multi-scale condition-guided diffusion module to reinforce the attribute-level visual representations.

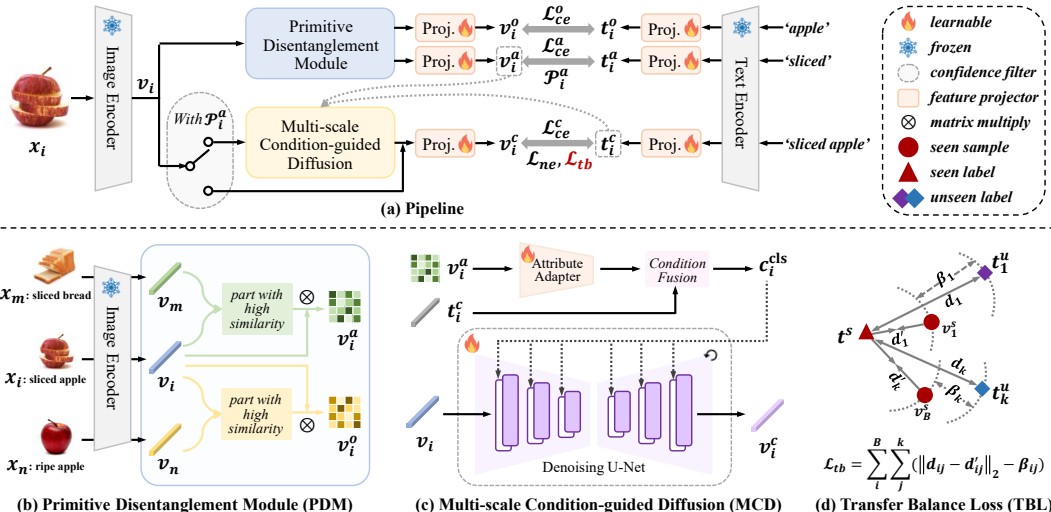

Figure 2: Illustration of our framework DETB, which consists of three main components: a Primitive Disentanglement Module (PDM); a proposed Multi-scale Condition-guided Diffusion (MCD) module that enhances attribute details by global semantic priors with localized visual disentangled representations; and a carefully designed class-adaptive Transfer Balancing Loss (TBL) that adjusts the semantic margins between seen and unseen compositions according to their inter-class similarity.

# 3 OUR APPROACH

## 3.1 TASK FORMULATION

OV-CZSL aims to enable the composition of both seen and novel attributes and objects. Each image $X$ is associated with two semantic labels: an attribute $A$ and an object $O$. We denote previously unseen concepts with an asterisk, such that unseen attributes and objects are represented as $A^*$ and $O^*$, respectively. The training set consists solely of observed attribute-object pairs, denoted by $Y^s = AO$. During inference, CZSL involves only novel attribute-object compositions with known primitives during training (i.e., $(AO)^*$), whereas OV-CZSL introduces more challenging cases involving unseen primitives. Specifically, the test set includes three types of novel compositions: (1) seen attribute with unseen object $AO^*$, (2) unseen attribute with seen object $A^*O$, and (3) both attribute and object unseen $A^*O^*$. The complete test label space is defined as $Y^u = (AO)^* \cup AO^* \cup A^*O \cup A^*O^*$. Importantly, there is no overlap between training and testing compositions, i.e., $Y^s \cap Y^u = \emptyset$.

## 3.2 PRELIMINARY

We propose a novel OV-CZSL framework named DETB, to achieve detail enhancement and transfer balance. Inspired by mainstream visual disentanglement approaches in CZSL, we adopt a three-branch architecture with dedicated disentanglers for separating attribute and object representations. Specifically, as illustrated in Figure 2, our proposed framework consists of three main components: a Primitive Disentanglement Module (PDM); a proposed Multi-scale Condition-guided Diffusion (MCD) module that enhances fine-grained attribute-specific details; and a designed class-adaptive Transfer Balancing Loss (TBL) designed to prevent overfitting to seen compositions.

## 3.3 PRIMITIVE DISENTANGLEMENT MODULE (PDM)

Followed by Saini et al. (2024), in the visual modality, for each sample $x_i$, we extract image features $v_i$ from the penultimate layer (prior to the average pooling operation) of a pre-trained ResNet18 He et al. (2016). In the semantic modality, embeddings of composition and primitives (i.e., $t_i^c$, $t_i^a$, and $t_i^o$) are extracted from BERT Devlin et al. (2019) separately.

Then, we introduce Primitive Disentanglement Module (PDM) that disentangles attribute and object features from image-level representations. In line with mainstream CZSL approaches, such as OADis

Saini et al. (2022), for each $x_i$, we select two assisted features randomly in the dataset: one with *the same attribute but a different object*, denoted as $x_m$, and another with *the same object but a different attribute*, denoted as $x_n$. The attribute disentangled feature $v_i^a$ is obtained by extracting the similarity weight from the $(x_i, x_m)$ pair and then multiplying it with $v_i$. Similarly, the object feature $v_i^o$ is extracted from $(x_i, x_n)$. Followed by Saini et al. (2024), in the visual modality, for each sample $x_i$, we extract image features $f_i$ from the penultimate layer (prior to the average pooling operation) of a pre-trained ResNet18 He et al. (2016). After aligning $v_i^a$ and $v_i^o$ to match the text embedding dimensions via MLP, the primitive classification probabilities can be calculated as:

$$\mathcal{P}_i^a = \frac{e^{\langle v_i^a, t_i^a \rangle / \tau}}{\sum_{y_i^a \in A} e^{\langle v_a, y_i^a \rangle / \tau}}, \mathcal{P}_i^o = \frac{e^{\langle v_i^o, t_i^o \rangle / \tau}}{\sum_{y_i^o \in O} e^{\langle v_o, y_i^o \rangle / \tau}}, \tag{1}$$

where $y_i^a$ and $y_i^o$ denote the attribute and object labels of $x_i$ respectively, $\tau$ is the temperature factor, and $\langle \cdot, \cdot \rangle$ stands for the cosine similarity between visual and semantic features.

### 3.4 MULTI-SCALE CONDITION-GUIDED DIFFUSION (MCD)

As attributes are fine-grained semantic concepts, while the global visual feature $v_i^c$ is natively object-centric, it often lacks sufficient detail to distinguish subtle attribute variations. To address this limitation, we draw inspiration from the powerful generative capacity of diffusion models, leveraging their ability to recover fine-grained details (*e.g.*, subtle textures, edge color variations, and material characteristics), to capture the key information necessary for correctly distinguishing between confusing attributes.

We identify hard samples as the primary bottleneck to generalization, where missing fine-grained details result in ambiguous attribute representations. Enhancing attribute representations of challenging samples is crucial for improving recognition accuracy. Therefore, we design a hard-aware filtering strategy that focuses on the most critical samples, avoiding redundant information as noise to easier instances and reducing computational cost.

Specifically, we identify hard samples based on attribute prediction errors from the PDM. Among misclassified instances, we rank confidence scores $\mathcal{P}_i^a$ and select the bottom $q\%$, forming a subset $\mathcal{H}$ for diffusion-based visual enhancement.

Then, the selected $\mathcal{H}$ are fed into our proposed Multi-scale Condition-guided Diffusion (MCD). Since the input $v_i$ is a high-dimensional feature represented as a 1D sequence, we introduce Unet1D and apply Gaussian noise following the standard forward process of diffusion:

$$v_{i,t} = \sqrt{\bar{\alpha}_t} \cdot v_i + \sqrt{1 - \bar{\alpha}_t} \cdot \epsilon, \quad \epsilon \sim \mathcal{N}(0, \mathbf{I}), \tag{2}$$

where $\bar{\alpha}_t = \prod_{s=1}^{t}(1 - \beta_s)$ denotes the cumulative product in the standard diffusion noise schedule. MCD is used to refine visual features via a class-guided denoising mechanism. Concretely, the class-specific condition $c_i^{\text{cls}}$ is formed by the disentangled attribute representation $v_i^a$ and the composition label embedding $t_i^c$:

$$c_i^{\text{cls}} = \delta \cdot Adapter(v_i^a) + (1 - \delta) \cdot t_i^c, \tag{3}$$

where $\delta$ is hyperparameter. Here we employ a learnable adapter to align the dimensions of $v_i^a$ and $t_i^c$, which is implemented using a two-layer MLP. As attribute representations are strongly influenced by the co-occurring object, the composition text $t_i^c$ constrains the attribute within its contextual environment, enabling $c_i^{\text{cls}}$ to incorporate both global contextual and locally disentangled guidance. We then predict noise under both conditions using the UNet1D:

$$\hat{\epsilon}_\theta = \epsilon_\theta(v_{i,t}, t, c_i^{\text{cls}}), \tag{4}$$

where $t$ is the timestep. This guided noise estimate can be used to recover a denoised representation:

$$\tilde{v}_i = \frac{1}{\sqrt{\bar{\alpha}_t}} \left( v_{i,t} - \sqrt{1 - \bar{\alpha}_t} \cdot \hat{\epsilon}_\theta \right). \tag{5}$$

The entire attribute enhancement process within MCD can be formulated as follows:

$$v_i^c = \begin{cases} \tilde{v}_i, & \text{if } i \in \mathcal{H} \\ v_i, & \text{otherwise} \end{cases}. \tag{6}$$

Subsequently, we constrain the predicted noise to match the input, which can be formulated as:

$$\mathcal{L}_{diff} = \mathbb{E}_{v_i, \epsilon, t, c_i^{\text{cls}}} [\, \|\epsilon - \hat{\epsilon}_\theta\|^2 \,]. \tag{7}$$

## 3.5 Class-adaptive Transfer Balance Loss (TBL)

Prior works have demonstrated that similarity-based consistency between seen and unseen classes promotes knowledge transfer. For instance, BSPC Saini et al. (2024) uses Neighborhood Expansion Loss (NEL) to propagate labels from seen concepts to semantically similar unseen ones. Intuitively, NEL pulls seen composition features closer to their most similar unseen neighbors.

However, this direct pulling operation may blur seen-unseen class boundaries. We think the model should learn an appropriate margin within similar seen-unseen pairs to better balance knowledge transfer and category discrimination. To this end, we propose a class-adaptive Transfer Balance Loss (TBL), which dynamically determines the inter-class distances during training.

Our key intuition is that similar pairs should be subject to a smaller margin compared to dissimilar ones. Therefore, we define the margin constraints based on the semantic similarity between class embeddings. For every sample $v_i^c$ in seen compositions, we retrieve the top-$k$ most similar unseen class embeddings $N_i^u = \{t_{i,1}^u, \cdots, t_{i,k}^u\}$, based on cosine similarity to $t_i^c$. Then, the class-adaptive margin $\beta_{ij}$ is defined as follows:

$$\beta_{ij} = \beta_{\max} \cdot \frac{e^{\langle t_i^c, t_{i,j}^u \rangle / \sigma}}{\sum_{l=1}^{k} e^{\langle t_i^c, t_{i,l}^u \rangle / \sigma}}, \tag{8}$$

where $\beta_{\max}$ is the upper bound of the margin, $\sigma$ is a temperature factor, and $\langle \cdot, \cdot \rangle$ stands for the cosine similarity.

For each seen-unseen pair, we finally define the triplet loss with margins as:

$$\mathcal{L}_{tb} = \frac{1}{B \cdot k} \sum_{i=1}^{B} \sum_{j=1}^{k} \left[ \left\| t_i^c - v_i^c \right\|_2 - \left\| t_i^c - t_{i,j}^u \right\|_2 - \beta_{ij} \right]_+, \tag{9}$$

where $t_{i,j}^u$ is the $j$-th most similar unseen class embedding for $t_i^c$, $B$ is batch-size, and the operator $[x]_+ = \max(0, x)$ denotes the standard hinge function.

## 3.6 Training Loss and Inference Phase

Similar to Eq. 1, once $v_i^c$ is obtained, it can be used to compute classification probabilities with compositional text features, which can be formally expressed as:

$$\mathcal{P}_i^c = \frac{e^{\langle v_i^c, t_i^c \rangle / \tau}}{\sum_{y_i^c \in Y^s} e^{\langle v_i^c, y_i^c \rangle / \tau}}, \tag{10}$$

where $y_i^c$ denotes the composition label of $x_i$, $\tau$ is the temperature factor, and $\langle \cdot, \cdot \rangle$ stands for the cosine similarity. Thus, the classification loss can be computed as follows:

$$\mathcal{L}_{ce}^n = -\frac{1}{|X_s|} \sum_{x_i \in X_s} \log \mathcal{P}_i^n, \ n \in \{a, o, c\}, \tag{11}$$

where $X_s$ is the training set. Following BSPC Saini et al. (2024), we also leverage NEL to facilitate knowledge transfer. The final loss is linearly combined as a whole, incorporating the above losses:

$$\mathcal{L} = \mathcal{L}_{ce}^c + \lambda_1 (\mathcal{L}_{ce}^a + \mathcal{L}_{ce}^o) + \mathcal{L}_{diff} + \lambda_2 \mathcal{L}_{tb} + \lambda_3 \mathcal{L}_{ne} \tag{12}$$

where $\lambda_1$, $\lambda_2$, and $\lambda_3$ are the weighting coefficients to balance the influence of each loss.

During the inference phase, consistent with mainstream approaches OADis Saini et al. (2022) and BSPC, assisted samples sharing the same primitives (i.e., $x_m$ and $x_n$) are unavailable. Consequently, the disentangled attribute feature $v_i^a$ cannot be derived, and the model performs prediction based solely on the holistic composition. Since the probability $\mathcal{P}_i^a$ is unavailable for filtering, we assume that all test samples require attribute enhancement through MCD (i.e., $v_i^c = \tilde{v}_i$), with the class-adaptive condition $c_i^{\text{cls}}$ set to empty. The final labels for the sample $x_i$ can be determined as follows:

$$\hat{c} = \arg\max_{c \in Y^u} \mathcal{P}_i^c. \tag{13}$$

Table 1: Dataset Splits on MIT-States, C-GQA, and VAW-CZSL. We denote $*$ for unseen primitives: $A$ and $O$ represent seen attributes and objects, while $A^*$ and $O^*$ denote unseen ones. Compositions are categorized as Seen pairs $AO$, Unseen pairs $(AO)^*$ with seen primitives and Unseen pairs $\{AO^*, A^*O, A^*O^*\}$ with unseen primitives.

| Datasets | Attributes | | Objects | | Training Set | Validation Set | Test Set |
|---|---|---|---|---|---|---|---|
| | $A$ | $A^*$ | $O$ | $O^*$ | $AO$ | $AO / (AO)^* / A^*O / AO^* / A^*O^*$ | $AO / (AO)^* / A^*O / AO^* / A^*O^*$ |
| MIT-states Isola et al. (2015) | 84 | 31 | 182 | 63 | 955 | 236 / 105 / 126 / 177 / 44 | 289 / 130 / 157 / 218 / 50 |
| C-GQA Naeem et al. (2021) | 311 | 102 | 504 | 170 | 4094 | 1012 / 447 / 525 / 517 / 147 | 1239 / 542 / 664 / 655 / 176 |
| VAW-CZSL Saini et al. (2022) | 330 | 135 | 406 | 110 | 7142 | 1767 / 803 / 1420 / 1253 / 412 | 2161 / 982 / 1737 / 1532 / 504 |

Table 2: Results (%) on MIT-States and C-GQA. We report Top-1 AUC, which balances between seen and unseen compositions with different bias terms. HM denotes the Harmonic Mean. Best accuracy values of different kinds of compositions $\{AO, (AO)^*, AO^*, A^*O, A^*O^*\}$ are also reported. AUC and HM are the most representative and stable metrics to evaluate the performance of models. The best and second-best results are marked in **bold** and underline, respectively.

| Methods | MIT-States | | | | | | | | | C-GQA | | | | | | | | |
|---|---|---|---|---|---|---|---|---|---|---|---|---|---|---|---|---|---|---|
| | AUC | HM | Seen | Unseen | $AO$ | $(AO)^*$ | $A^*O$ | $AO^*$ | $A^*O^*$ | AUC | HM | Seen | Unseen | $AO$ | $(AO)^*$ | $A^*O$ | $AO^*$ | $A^*O^*$ |
| LE Nagarajan & Grauman (2018) | 1.01 | 7.64 | 16.29 | 9.46 | 10.24 | 11.38 | 5.98 | 4.15 | 2.87 | 1.17 | 8.39 | 19.37 | 8.36 | 10.76 | 6.51 | 9.53 | 2.67 | 1.08 |
| CompCos Mancini et al. (2021) | 1.97 | 10.22 | 26.53 | 10.29 | 14.32 | 21.09 | 5.86 | 2.89 | 0.63 | 2.35 | 9.64 | 40.19 | 7.25 | 21.19 | 20.24 | 4.47 | 1.95 | 0.26 |
| OADis Saini et al. (2022) | 1.83 | 9.55 | 25.35 | 10.79 | 12.68 | 16.06 | 6.40 | 5.41 | 1.34 | 2.33 | 9.74 | **42.88** | 7.12 | 20.86 | 15.19 | 6.17 | 3.47 | 0.61 |
| SCEN Li et al. (2022) | 1.73 | 9.72 | 22.08 | 8.25 | 11.85 | **30.02** | 3.82 | 0.33 | 0.08 | 1.97 | 9.03 | 41.65 | 7.83 | 20.65 | 21.42 | 3.61 | 1.08 | 0.05 |
| CANet Wang et al. (2023b) | 2.40 | 10.52 | 26.42 | 9.54 | 16.56 | 23.08 | 6.15 | 4.08 | 0.58 | 3.04 | 11.96 | 40.52 | 9.21 | 22.43 | 20.87 | 4.95 | 2.03 | 0.64 |
| BSPC Saini et al. (2024) | 2.41 | 10.94 | 29.02 | 11.13 | 14.11 | 18.87 | 8.24 | 5.49 | 3.54 | 3.18 | 12.11 | 42.38 | 9.77 | 19.78 | 16.07 | 12.86 | 2.87 | 3.04 |
| **DETB (Ours)** | **2.45** | **11.12** | **30.45** | **11.65** | **16.85** | 19.94 | **8.89** | **6.74** | **3.82** | **3.78** | **14.16** | 42.73 | **13.27** | 22.19 | 16.82 | **16.27** | **4.28** | **3.81** |

# 4 EXPERIMENTS

## 4.1 EXPERIMENTAL SETUP

**Datasets.** We evaluate our model on three benchmark datasets: 1) *MIT-States* Isola et al. (2015) contains diverse real-world objects (*e.g.*, cheese, sea) described by attributes (*e.g.*, molten, dark). 2) *C-GQA* Naeem et al. (2021) is the most extensive CZSL dataset, which is newly created based on the Stanford GQA dataset Hudson & Manning (2019) for VQA tasks, composed of attributes (*e.g.*, red, dirty) and objects (*e.g.*, pen, window) commonly found in daily life. 3) *VAW-CZSL* Saini et al. (2022) is a large-scale dataset derived from the VAW (Visual Attributes in the Wild) dataset Pham et al. (2021), composed of attributes (*e.g.*, furry, wet) and objects (*e.g.*, dog, umbrella) grounded in real-world images. Its long-tailed distribution and filtered compositions make it well-suited for open-vocabulary scenarios. The split details are shown in Table 1.

**Evaluation Metrics.** Since the validation and test set include both seen and unseen compositions, CZSL models inevitably exhibit a bias towards seen ones. Following the Generalized CZSL evaluation protocol proposed by Purushwalkam et al. (2019), we apply a scalar bias to calibrate predictions. Due to the difficulty of OV-CZSL, we adopt a closed-world evaluation, computing metrics only on valid unseen compositions. We vary the scalar to plot an unseen-seen accuracy curve (seen on X-axis, unseen on Y-axis) and calculate the Area Under Curve (AUC). The best Harmonic Mean (HM) is also reported to assess bias balance. Best accuracy values are also reported for Seen $AO$, Unseen pairs$\{(AO)^*\}$ with seen primitives, and Unseen pairs$\{AO^*, A^*O, A^*O^*\}$. Among these metrics, AUC and HM are the most representative and stable metrics to evaluate the performance of models.

## 4.2 COMPARISON WITH STATE-OF-THE-ARTS

**Compared Methods.** Currently, BSPC Saini et al. (2024) is the only existing method strictly under the OV-CZSL setting. Therefore, we include a range of CZSL methods for comparison. Among them, OADis Saini et al. (2022) is the most similar to our approach. We also compare with recent models, such as SCEN Li et al. (2022) and CANet Wang et al. (2023b). To ensure a fair comparison, all baselines use visual features from ResNet18 and word embeddings from BERT.

**Results on MIT-States and C-GQA.** As shown in Table 2, our proposed DETB achieves state-of-the-art performance on both the MIT-States and C-GQA datasets, with significant improvements across most evaluation metrics. It demonstrates a well-balanced prediction between seen and unseen categories, further validating the effectiveness of our method. Notably, our model achieves

particularly evident gains on unseen pairs with unseen primitives (*i.e.*, $AO^*$, $A^*O$, $A^*O^*$), which directly confirms the effectiveness of our approach in alleviating overfitting to seen pairs. Although DETB performs favorably on most metrics, it shows relatively weaker performance on unseen pairs with seen primitives (*i.e.*, $(AO)^*$), where it still falls short compared to some traditional CZSL methods. It's likely because $(AO)^*$ emphasizes modeling attribute–object relations, whereas DETB prioritizes primitive knowledge transfer and fine-grained visual enhancement; explicitly capturing intra-compositional relations is not a primary strength of its design.

**Results on VAW-CZSL.** VAW-CZSL is derived from the multi-label VAW dataset, where the least frequent label is assigned to each image. Consequently, the top-1 prediction may capture attributes present but not labeled. Following prior CZSL methods, we report evaluation metrics based on the top-3 predictions. As shown in Table 3, compared with other traditional CZSL methods, CANet performs better on $AO$, while SCEN excels on $(AO)^*$. In contrast, DETB achieves the best re-

Table 3: Results (%) on VAW-CZSL. All metrics are shown in Top-3 predictions. AUC and HM are the most representative and stable metrics to evaluate the performance of models. The best and second-best results are marked in **bold** and underline, respectively.

| Methods | AUC | HM | $AO$ | $(AO)^*$ | $A^*O$ | $AO^*$ | $A^*O^*$ |
|---|---|---|---|---|---|---|---|
| LE Nagarajan & Grauman (2018) | 1.49 | 8.27 | 15.62 | 10.48 | 5.79 | 2.78 | 0.98 |
| CompCos Mancini et al. (2021) | 2.69 | 10.68 | 20.21 | 20.58 | 5.04 | 2.48 | 0.50 |
| OADis Saini et al. (2022) | 2.68 | 10.91 | 21.19 | 15.65 | 6.75 | 3.16 | 0.76 |
| SCEN Li et al. (2022) | 2.53 | 10.64 | 19.06 | **20.76** | 4.52 | 2.05 | 0.42 |
| CANet Wang et al. (2023b) | 2.89 | 11.21 | **24.56** | 18.42 | 5.74 | 2.86 | 0.95 |
| BSPC Saini et al. (2024) | 2.91 | 11.35 | 23.02 | 16.18 | 7.86 | 3.37 | 1.36 |
| **DETB (Ours)** | **3.15** | **11.99** | 23.76 | 17.70 | **8.61** | **4.15** | **1.38** |

sults across other metrics, particularly on unseen compositions with unseen primitives (*i.e.*, $AO^*$, $A^*O$, $A^*O^*$), highlighting superior generalization to novel attribute-object pairs.

## 4.3 ABLATION ANALYSIS

**Effect of MCD and TBL.** We evaluate the effectiveness of Multi-scale Condition-guided Diffusion (MCD) and Transfer Balance Loss (TBL), and report the results of ablation studies on the MIT-States dataset in Table 4. Experimental results of the three variants we designed indicate that, removing either MCD or TBL from the full DETB framework results in a notable performance drop. Specifically, the introduction of MCD brings greater improvements for compositions involving seen primitives, especially for $AO$ pairs. It clearly highlights the benefits of enhancing visual details in global features for recognizing seen primitives. Moreover, leveraging TBL substantially boosts recognition performance across various unseen pairs, verifying its strong generalization ability under open-vocabulary settings.

Table 4: Results (%) on MIT-States, C-GQA, and VAW-CZSL w/ or w/o MCD and TBL. For MIT-States and C-GQA, we report performance based on Top-1 prediction; for VAW-CZSL, we report Top-3.

| Datasets | MCD | TBL | AUC | HM | $AO$ | $(AO)^*$ | $A^*O$ | $AO^*$ | $A^*O^*$ |
|---|---|---|---|---|---|---|---|---|---|
| MIT-States | | | 2.41 | 10.94 | 14.11 | 18.87 | 8.24 | 5.49 | 3.54 |
| | ✓ | | 2.42 | 11.01 | 15.89 | 19.48 | 8.74 | 5.52 | 3.56 |
| | | ✓ | 2.43 | 11.07 | 16.45 | 19.02 | 8.32 | 6.44 | 3.71 |
| | ✓ | ✓ | **2.45** | **11.12** | **16.85** | **19.94** | **8.89** | **6.74** | **3.82** |
| C-GQA | | | 3.18 | 12.11 | 19.78 | 16.07 | 12.86 | 2.87 | 3.04 |
| | ✓ | | 3.55 | 14.07 | 21.19 | 16.22 | 14.99 | 3.39 | 3.16 |
| | | ✓ | 3.53 | 13.83 | **22.58** | 14.92 | 15.63 | 4.01 | 3.34 |
| | ✓ | ✓ | **3.78** | **14.16** | 22.19 | **16.82** | **16.27** | **4.28** | **3.81** |
| VAW-CZSL | | | 2.91 | 11.35 | 23.02 | 16.18 | 7.86 | 3.37 | 1.36 |
| | ✓ | | 3.06 | 11.57 | 23.58 | 17.56 | **8.62** | 3.81 | 1.19 |
| | | ✓ | 3.13 | 11.87 | 23.16 | 17.30 | 8.46 | **4.34** | **1.39** |
| | ✓ | ✓ | **3.15** | **11.99** | **23.76** | 17.70 | 8.61 | 4.15 | 1.38 |

**Effect of Multi-scale Condition Fusion in MCD.** We evaluate the different values of the condition fusion weight $\delta$ in MCD, and report the results of ablation studies on the MIT-States dataset in Table 5. Specifically, $\delta$ controls the fusion ratio between the disentangled attribute representation $v_i^a$ and the compositional text embedding $t_i^c$. When $\delta = 0$, MCD is guided solely by $t_i^c$; when $\delta = 1$, it is guided only by $v_i^a$. Experimental results of the five variants we designed indicate that appropriately combining semantic and visual information leads to

Table 5: Results (%) on MIT-States with different scale guided in the MCD module. $\delta$ is the hyperparameter in Eq. 3.

| Scales $\delta$ | AUC | HM | $AO$ | $(AO)^*$ | $A^*O$ | $AO^*$ | $A^*O^*$ |
|---|---|---|---|---|---|---|---|
| 0.00 | 2.43 | 10.96 | 15.84 | 19.74 | 8.59 | 6.61 | 3.45 |
| 0.25 | 2.41 | 10.89 | **16.89** | 18.96 | 8.40 | 6.67 | 3.67 |
| 0.50 | **2.45** | **11.12** | 16.85 | **19.94** | **8.89** | 6.74 | **3.82** |
| 0.75 | 2.40 | 10.73 | 16.40 | 19.66 | 8.28 | 6.80 | 3.64 |
| 1.00 | 2.43 | 10.95 | 15.65 | 19.87 | 8.50 | **6.96** | 3.62 |

better overall performance. Moreover, fully relying on visual attribute guidance (*i.e.*, $\delta = 1$) results in a decline in overall performance, indicating that dependence on a single modality limits the model's generalization capability. However, its best results on unseen compositions with seen attributes (*i.e.*, $(AO)^*$ and $AO^*$), indirectly demonstrate the advantage of enhanced attribute-level features for recognizing seen attributes.

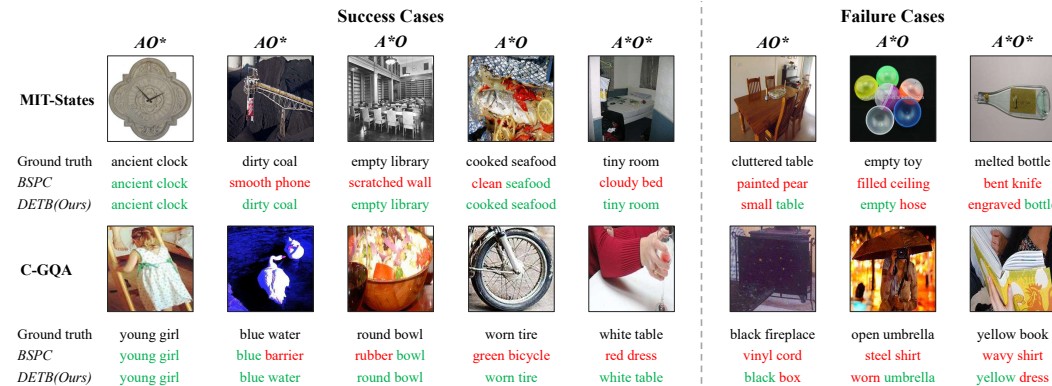

Figure 3: Qualitative comparison with BSPC Saini et al. (2024). We present predictions on randomly selected cases from MIT-States and C-GQA, focusing on unseen pairs with unseen primitives (*i.e.*, $AO^*$, $A^*O$, $A^*O^*$). Green / Red denotes the correct / wrong predictions.

### 4.4 QUALITATIVE RESULTS

**Performance comparison.** Since OV-CZSL is more challenging than traditional CZSL due to the inclusion of unseen primitives during testing, we select samples from MIT-States and C-GQA belonging to $\{AO^*, A^*O, \text{and } A^*O^*\}$, and compare our predictions with BSPC Saini et al. (2024) in Figure 3. Our DETB generates more accurate and semantically coherent predictions, especially for samples in novel scenarios $A^*O^*$. It demonstrates more robust generalization to unseen pairs and supports the effectiveness of our proposed TBL. Failure cases (*e.g.*, 8th example in C-GQA) often stem from semantic entanglement, which hinders accurate prediction. BSPC's misclassification underscores the limitation of relying solely on global visual features in complex contexts. In contrast, DETB accurately predicts the attribute 'yellow', validating the effectiveness of our attribute enhancement design in MCD.

**T-SNE visualization of hardest samples fed into MCD.** To directly validate the effectiveness of MCD in enriching the discriminative features of samples, we visualize the feature distributions before and after applying MCD using t-SNE. As shown in Figure 4, the sample points belonging to the same class in (b) are more compact compared to (a) (*e.g.*, 'weathered fence'), indicating that the MCD module enhances intra-class consistency, which makes samples in the same class more tightly clustered in the feature space. Meanwhile, the class boundaries in (b) are more distinct, and fine-grained categories become more separable (*e.g.*, confusing pairs 'murky soup' and 'thick soup'), demonstrating the effectiveness of MCD in improving fine-grained discriminability.

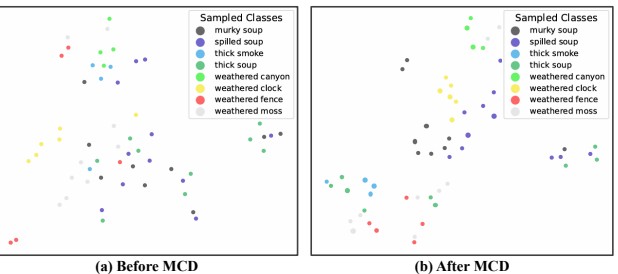

Figure 4: T-SNE visualization of the hardest samples from MIT-States before and after enhancement by MCD.

Please refer to the Appendix for Implementation Details A, more Ablations B, and Discussion C.

### 5 CONCLUSION

We propose a novel OV-CZSL framework named DETB, which simultaneously achieves fine-grained detail enhancement and knowledge transfer balance. We design the MCD module to refine attribute representations of challenging samples by leveraging both global context and local disentangled features. In addition, our class-adaptive TBL dynamically adjusts decision boundaries based on semantic similarity, boosting DETB's generalization to unseen compositions. Extensive quantitative and qualitative results demonstrate that DETB consistently outperforms existing approaches.

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

## A  IMPLEMENTATION DETAILS

We implement DETB based on Pytorch 1.12.1. The model is trained and evaluated on NVIDIA GeForce RTX 3090 for all three datasets. For fair comparison with existing CZSL Saini et al. (2022); Li et al. (2022); Wang et al. (2023b) and OV-CZSL Saini et al. (2024) methods, we use image features extracted from a frozen ResNet18 He et al. (2016) pre-trained on ImageNet Deng et al. (2009) without finetuning. We use BERT Devlin et al. (2019) text embeddings for labels. Following OADis Saini et al. (2022), we use image augmentations (random crop, horizontal flip) for our method. All projectors are implemented as trainable two-layer MLPs. In the PDM module, $\tau$ is always set to 0.05. In the MCD module, the inference steps $t$ are set to 1000 across all three datasets. The confidence filter threshold $q\%$ is set to 10% for all three datasets. In TBL, we configure the number of neighbors $k$ as 5, 5, 10 for MIT-States Isola et al. (2015), C-GQA Naeem et al. (2021), and VAW-CZSL Saini et al. (2022), separately. $\beta_{max}$ and $\sigma$ are always set to 1 and 10, respectively. In the final loss $\mathcal{L}$, $\lambda_1$ is 0.05, $\lambda_2$ is 0.3, 0.2, and 0.5 for MIT-States, C-GQA, and VAW-CZSL, separately. $\lambda_3$ Consistent with the setting in BSPC Saini et al. (2024), $\lambda_3$ is fixed to 0.8 across all datasets. Our optimization setup uses Adam optimizer with a learning rate of $5 \times 10^{-5}$ for MIT-States and VAW-CZSL, and $1 \times 10^{-4}$ for C-GQA.

## B  ABLATION ANALYSIS

Table 6: Results (%) on MIT-States with different filter ratios in the MCD module. We select the bottom $q\%$ of samples with the lowest attribute prediction confidence $\mathcal{P}_i^a$ and feed them into the MCD module.

| Filter Ratio $q\%$ | AUC | HM | $AO$ | $(AO)^*$ | $A^*O$ | $AO^*$ | $A^*O^*$ |
|---|---|---|---|---|---|---|---|
| 5% | 2.43 | 10.96 | 16.68 | 19.60 | 8.28 | 6.59 | 3.60 |
| 10% | **2.45** | **11.12** | **16.85** | 19.94 | **8.89** | 6.74 | **3.82** |
| 20% | 2.40 | 10.89 | 15.52 | **20.11** | 8.05 | 6.83 | 3.45 |
| 30% | 2.36 | 10.87 | 16.56 | 18.65 | 7.82 | 6.88 | 3.39 |
| 40% | 2.33 | 10.65 | 15.15 | 18.20 | 7.54 | **7.25** | 3.54 |

Table 7: Results (%) on MIT-States with different numbers of neighbors in TBL.

| Neighbors Num $k$ | AUC | HM | $AO$ | $(AO)^*$ | $A^*O$ | $AO^*$ | $A^*O^*$ |
|---|---|---|---|---|---|---|---|
| 1 | 2.36 | 10.80 | 15.50 | 19.60 | 8.56 | 6.52 | 3.29 |
| 3 | 2.42 | 11.00 | 15.62 | 19.96 | 8.63 | 6.71 | 3.47 |
| 5 | **2.45** | **11.12** | **16.85** | 19.94 | **8.89** | 6.74 | **3.82** |
| 7 | 2.38 | 10.89 | 15.85 | **20.17** | 8.49 | **6.76** | 3.64 |
| 10 | 2.37 | 10.87 | 16.25 | 18.87 | 7.56 | 6.59 | 3.32 |

**Effect of Filter by Attribute Confidence in MCD.** We evaluate the different values of filter ratios $q\%$ in MCD, and report the results of ablation studies on the MIT-States dataset in Table 6. Specifically, we forward the bottom $q\%$ of samples ranked by attribute prediction confidence $\mathcal{P}_i^a$ into MCD to enhance attribute awareness. Experimental results show that a moderate filtering ratio improves overall model performance. In particular, when $q\% = 10\%$, the model achieves optimal AUC and HM scores. Although gradually increasing $q$ leads to a consistent accuracy gain on $AO^*$, further confirming the benefit of MCD in recognizing seen attributes, we do not advocate using a large $q$. First, it incurs higher computational cost; second, the overall performance (*i.e.*, AUC and HM) deteriorates, potentially due to over-enhancement of features, which may introduce noise and hinder generalization from seen to unseen compositions. In summary, the results confirm the effectiveness of the confidence-driven filtering mechanism. An appropriate selection ratio allows for targeted enhancement of challenging samples, leading to improved compositional generalization.

**Effect of the Number of Neighbors in TBL.** We evaluate the different values of neighbor number $k$ in TBL, and report the results of ablation studies on the MIT-States dataset in Table 7. The results indicate that a proper selection of $k$ is crucial for enhancing compositional generalization. When $k = 5$, our model achieves optimal performance across multiple metrics. It suggests that moderately expanding the neighborhood during semantic transfer is beneficial, which helps estimate the category semantics more robustly and reduces the risk of overfitting to a single class. However, as $k$ increases further (*e.g.*, $k = 10$), the model's performance degrades, possibly due to the introduction

Table 8: Results (%) on MIT-States with different TBL weights $\lambda_2$ in the final loss $\mathcal{L}$.

| TBL weights $\lambda_2$ | AUC | HM | $AO$ | $(AO)^*$ | $A^*O$ | $AO^*$ | $A^*O^*$ |
|---|---|---|---|---|---|---|---|
| 0.1 | 2.44 | 11.06 | 16.69 | **20.77** | 8.10 | 6.42 | 3.26 |
| 0.2 | 2.42 | 11.02 | 16.29 | 19.48 | 8.40 | 6.42 | 3.60 |
| 0.3 | **2.45** | **11.12** | 16.85 | 19.94 | **8.89** | 6.74 | **3.82** |
| 0.4 | 2.31 | 10.69 | **17.18** | 18.03 | 7.68 | **6.82** | 3.36 |
| 0.5 | 2.29 | 10.73 | 15.40 | 17.07 | 7.56 | 6.57 | 3.29 |

of noisy neighbors that weaken the discriminative power during class transfer. Overall, a moderate neighborhood size offers a good trade-off between semantic generalization and class distinctiveness.

**Effect of TBL Weight in the Final Loss.** We evaluate the different values of TBL weight $\lambda_2$ in $\mathcal{L}$, and report the results of ablation studies on the MIT-States dataset in Table 8. The results indicate that a proper balance between the TBL component and the overall loss is essential for optimal compositional generalization. When $\lambda_2 = 0.3$, the model achieves the best performance across multiple metrics, suggesting that moderately reinforcing TBL positively contributes to modeling unseen compositions. However, with a larger weight (*e.g.*, $\lambda_2 = 0.5$), the model's performance drops, indicating that overemphasizing TBL may suppress the optimization of other crucial learning signals. In summary, assigning TBL a moderate weight helps achieve better balance and generalization across different composition splits.

## C  DISCUSSION

**Why not directly generate unseen samples using diffusion models?** As illustrated in Table 1 1 in our manuscript, the OV-CZSL setting involves approximately 6,000 unseen classes during validation, and around 7,000 unseen classes during testing in large-scale datasets like VAW-CZSL. Using MCD on our training GPUs, each image takes roughly 1 second by sampling, which greatly slows down training and increases computational overhead. Moreover, many unseen classes are visually similar, particularly those sharing the same object but differing in attributes. Thus, to enable effective generation of such fine-grained unseen samples, future work would require stronger and more discriminative conditions to guide the diffusion process.

**Why is TBL not applied to the attribute branch?** TBL defines a class-adaptive margin $\beta$ based on semantic similarity between seen and unseen classes. However, as fine-grained semantic concepts, attributes often exhibit inconsistency between their semantic and visual similarities. In CZSL datasets, attributes are grouped into categories like color (*e.g.*, white, red), size (*e.g.*, big, small), or state (*e.g.*, clean, dirty). While attributes within a category are semantically close, their visual appearances can vary drastically (*e.g.*, 'black' and 'white'). Therefore, applying a semantic similarity–based margin constraint in TBL is not suitable for knowledge transfer between seen and unseen attributes. It can easily introduce noise, so we do not apply TBL to the attribute branch.

**Use of VLMs?** Recent CZSL works increasingly incorporate Vision-Language Models (VLMs) like CLIP Radford et al. (2021) to project data into a shared semantic space. Due to the fine-grained nature of attributes as semantic concepts, we consider that even high-quality global visual features obtained from VLMs still suffer from a lack of attribute details. In future work, we plan to leverage their strong zero-shot capabilities to further improve CZSL performance under the open-vocabulary setting.

## D  LLM USAGE STATEMENT

We only used GPT to polish the content of our manuscript and did not use LLM for any other purpose.

