# OpenReview forum: "Detail Enhancement and Transfer Balance for Open-Vocabulary Compositional Zero-Shot Learning"
_ICLR.cc/2026/Conference — ICLR 2026 Conference Withdrawn Submission_

### Official Review · Reviewer_o89D · 2025-10-27

**Soundness:** 3
**Presentation:** 3
**Contribution:** 2
**Rating:** 6
**Confidence:** 3

**Summary:**

This paper proposes a novel framework, Detail Enhancement and Transfer Balance, to address the challenges of Open-Vocabulary Compositional Zero-Shot Learning (OV-CZSL), where unseen attribute–object compositions, including novel attributes or objects, must be recognized. The method tackles two key issues—insufficient fine-grained attribute representation and indiscriminate knowledge transfer—by introducing a Multi-scale Condition-guided Diffusion (MCD) module that refines visual features through global-local integration, and a Transfer Balance Loss (TBL) that adaptively regulates semantic margins to balance transfer and separation between seen and unseen compositions. Experiments on three benchmark datasets demonstrate that DETB consistently outperforms prior approaches and achieves state-of-the-art performance.

**Strengths:**

1.This work leverages diffusion models to enhance fine-grained attribute representation in a discriminative learning setting, proving to be a novel and effective strategy.
2.This work proposes the Transfer Balance Loss, a novel solution to the critical issue of balancing knowledge transfer and discrimination for effective compositional generalization.
3.Extensive evaluation on three benchmarks shows the method significantly outperforms existing alternatives, confirming its effectiveness.

**Weaknesses:**

1. The model focuses on attribute enhancement but lacks explicit modeling of attribute–object relations, which might limit compositional reasoning in complex visual contexts.
2. While DETB achieves strong overall results, its performance on certain metrics or composition types shows noticeable limitations.

**Questions:**

1. Details on the computational cost and scaling potential of the diffusion module are currently lacking. Specifically, the paper does not report training time (e.g., per epoch or total), inference speed per image, or the additional memory and computational overhead.

---

### Official Review · Reviewer_Dpq3 · 2025-10-30

**Soundness:** 3
**Presentation:** 3
**Contribution:** 2
**Rating:** 4
**Confidence:** 4

**Summary:**

This paper introduces DETB, a new framework for Open-Vocabulary Compositional Zero-Shot Learning (OV-CZSL). The method tackles two key challenges: the lack of fine-grained detail in global features for attribute recognition and the imbalanced knowledge transfer between seen and unseen compositions. The authors propose a diffusion-based module (MCD) to enhance attribute details in hard samples and a class-adaptive Transfer Balance Loss (TBL) to create clearer class boundaries. Experiments on three public datasets show that DETB achieves state-of-the-art performance.

**Strengths:**

This paper introduce a new OV-CZSL framework, DETB, that creatively utilizes the strong generative ability of diffusion models for fine-grained detail recovery, directly addressing the critical challenge of attribute feature enhancement. The method of injecting external knowledge is a promising research direction for the CZSL field. Meanwhile, the authors provide empirical evidence across multiple benchmarks to demonstrate that this strategy is not just a conceptual novelty but is highly effective in practice, achieving good results by successfully balancing detail enhancement and knowledge transfer.

**Weaknesses:**

1. The paper's justification for employing a diffusion model, while novel, feels incomplete. The core motivation is to recover fine-grained details by leveraging the model's generative priors. However, this introduces significant computational and architectural complexity. It is not sufficiently clear why such a heavyweight solution is necessary for what is ultimately a feature enhancement task. The authors do not provide a comparative analysis against simpler, more efficient feature refinement techniques (e.g., attention-based feature re-weighting, dedicated residual refinement blocks) that could potentially achieve similar goals without the substantial overhead of an iterative denoising process. The paper needs to better demonstrate that the unique properties of a diffusion model are essential here, beyond just its general power as a generative tool.

2. I also found the paper's limited engagement with  VLMs to be a significant weakness, especially since a key part of the authors' narrative is introducing external knowledge.  The most direct and powerful source of external knowledge is arguably CLIP and its successors. Is the complex diffusion module a fundamentally superior way to get attribute details, or is it simply a very elaborate workaround for the limitations of a dated ResNet-18 backbone? I can't shake the feeling that much of the "lack of detail" issue could have been addressed more directly and efficiently by simply using a stronger VLM feature extractor from the start. To propose one method of introducing external knowledge (diffusion) while ignoring the most standard VLMs and relegating it to future work feels like a major oversight that sidesteps a crucial baseline comparison.

**Questions:**

1. I have some significant concerns regarding the computational cost of the MCD module. The note in Line 727 that inference on a single sample takes roughly one second, which is a massive overhead for a recognition task. Could you please provide a more direct, quantitative comparison of the total training and testing times of your full DETB framework against a key baseline like BSPC, or at least against your own model without MCD? This would be critical for understanding the practical trade-offs of your approach.
2. Following up on the cost, I'm not entirely convinced that the performance gain from MCD justifies its inclusion. Looking at your ablation in Table 4 for MIT-States, the model with just TBL achieves an AUC of 2.43, while the full model with both MCD and TBL reaches 2.45. While this is an improvement, it seems quite marginal for such an expensive component. Could you elaborate on why this modest gain is worth the substantial computational cost? Is there a particular subset of challenging compositions where MCD provides a much larger, more significant boost that isn't fully captured by the overall metrics?
3. To really solidify the contribution of using a diffusion model, I would strongly recommend including comparisons against alternative, more lightweight feature enhancement modules. For instance, how would a simple residual block or an attention block designed to refine the attribute features perform in place of MCD? This would help clarify if MCD offers a unique advantage or if it's primarily making up for the deficiencies of an older backbone.

---

### Official Review · Reviewer_iTng · 2025-10-31

**Soundness:** 3
**Presentation:** 2
**Contribution:** 2
**Rating:** 4
**Confidence:** 4

**Summary:**

This paper addresses two challenges in OV-CZSL: the lack of fine-grained attribute details in global visual features and the blurred class boundaries caused by indiscriminate knowledge transfer. The authors propose a new framework, DETB, which integrates a Multi-scale Condition-guided Diffusion module to enhance attribute-specific details for hard samples and a Transfer Balance Loss that introduces class-adaptive margins to balance transferability and discriminative ability. Experiments conducted on three standard benchmarks demonstrate that DETB achieves superior performance compared to existing methods across most metrics.

**Strengths:**

1) The paper clearly identifies two central challenges in OV-CZSL, and introduce diffusion models for fine-grained visual detail enhancement. The idea provides new insights for compositional generalization.

2) The method proposed in the paper is novel to me. The MCD module selectively enhances hard samples by fusing semantic and visual conditions, effectively addressing the issue of missing attribute details. Meanwhile, the TBL dynamically adjusts the margin between seen and unseen compositions based on semantic similarity, which alleviates over-transfer and improves the balance between generalization and discrimination.

3) Comprehensive empirical validation. Extensive experiments are conducted on three benchmark datasets with detailed ablation studies.

**Weaknesses:**

### Weaknesses

1) Limited theoretical justification and weak methodological motivation. The paper lacks a deeper theoretical or statistical analysis of how MCD enhances feature separability in the embedding space. The arguments rely primarily on empirical observations and qualitative visualization. Although the authors claim that diffusion models possess detail recovery capability, they do not provide a systematic explanation of why diffusion is inherently superior to other generative paradigms for fine-grained attribute enhancement.

2) Theoretical foundation and methodological motivation should be further strengthened. Although introducing diffusion models into the CZSL domain is novel, the paper lacks deeper theoretical or statistical analysis explaining how MCD enhances feature separability in the embedding space. The arguments rely primarily on empirical observations and qualitative visualization. While the authors claim that diffusion models possess detail recovery capability, they do not provide a systematic explanation of why diffusion is inherently superior to other paradigms for fine-grained attribute enhancement.

3) Inconsistent inference-time strategy. During inference, since auxiliary samples are unavailable, the model applies MCD enhancement to all test samples with an empty class-conditioning input. This contradicts the training strategy, where only hard samples are enhanced. Could such a discrepancy between training and inference introduce unnecessary noise for already easy samples?

4) Limited improvement in specific metrics. While DETB achieves overall sota results, its performance in  CZSL settings, particularly on the unseen composition set with seen primitives (AO)*, is weaker than some prior methods. This indicates that DETB tends to favor open-vocabulary generalization but may sacrifice compositional generalization, which limits its applicability to broader CZSL scenarios.

**Questions:**

See the weakness.

---

### Official Review · Reviewer_pEsW · 2025-11-01

**Soundness:** 2
**Presentation:** 3
**Contribution:** 2
**Rating:** 4
**Confidence:** 4

**Summary:**

This paper tackles the challenging task of Open-Vocabulary Compositional Zero-Shot Learning (OV-CZSL). The authors identify two key problems with existing methods: (1) global features, which are often object-centric, lack the fine-grained visual details needed to distinguish attributes, and (2) simplistic knowledge transfer mechanisms (like aligning seen and unseen class embeddings) can blur decision boundaries and cause overfitting to seen compositions. The proposed framework is built on a standard three-branch (attribute, object, composition) architecture. Experiments on three OV-CZSL benchmarks (MIT-States, C-GQA, VAW-CZSL) show that DETB achieves state-of-the-art results, particularly on the main AUC and Harmonic Mean (HM) metrics.

**Strengths:**

Novel Application of Diffusion Models: The use of a diffusion model (MCD) as a feature refiner for a discriminative task (classification) is a novel and interesting idea. Applying it selectively to "hard" samples (identified by low attribute confidence) is a sensible and efficient approach. The t-SNE in Fig. 4 provides some evidence that this module helps in creating more compact and separable clusters.

Well-Motivated Loss Function: The Transfer Balance Loss (TBL) is well-motivated. The core problem of CZSL is balancing generalization with discrimination, and the paper correctly identifies that naive alignment can be harmful. The idea of a dynamic, similarity-based margin (Eq. 8) is an elegant solution that directly addresses this trade-off, allowing transfer for close concepts while enforcing separation for more distant ones.

Good Ablation Studies: The paper includes a comprehensive set of ablations that validate the design choices.

**Weaknesses:**

My primary concerns revolve around the clarity and consistency of the MCD module's operation, particularly the discrepancy between training and inference.

1. Clarity on MCD Architecture (Q1): The MCD module is the most novel part, but its implementation details are somewhat underspecified. The paper describes $v_i$ as being "from the penultimate layer (prior to the average pooling operation)" (Sec. 3.3), which is typically a 2D spatial feature map (e.g., HxWxC). However, Sec 3.4 describes $v_i$ as a "1D sequence" and applies a "Unet1D". This is contradictory.

Q1: Please clarify the exact architecture. Is the 2D feature map first flattened into a 1D sequence before being fed to the Unet1D? Or is the "penultimate layer" referring to the 1D feature after pooling? A clear description of the input/output shapes and the "Unet1D" architecture is needed.

2. Contradiction in Inference Procedure (Q2 & Q3): There appears to be a critical contradiction in the paper regarding the inference process.Sec 3.4 states that MCD is conditioned on $c_{i}^{cls}$, which requires the disentangled visual attribute $v_i^a$ (Eq. 3).Sec 3.6 states that at inference, assisted samples are "unavailable," and therefore the disentangled attribute feature $v_i^a$ "cannot be derived."Sec 3.6 then states that at inference, all samples are passed through MCD, but the condition $c_{i}^{cls}$ is "set to empty."

Q2: If the condition is "empty" (unconditional diffusion), how can the model possibly enhance attribute-specific details? The entire motivation is that the diffusion is guided by the attribute. An unconditional model would have no basis to refine "dirty" vs. "clean." This seems to fundamentally undermine the proposed mechanism.

Q3: Is it possible that $v_i^a$ is derived at test time, perhaps using a different, non-assisted method? Or is the $c_{i}^{cls}$ at test time only the text embedding $t_i^c$ (i.e., $\delta=0$)? Please clarify this procedure, as the current description seems logically inconsistent and contradicts the core motivation of the MCD module.

3. Disentanglement Module (PDM) as a Black Box: The paper relies heavily on a "Primitive Disentanglement Module" (PDM, Sec 3.3) which is only briefly described and cited. This module requires sampling other images ($x_m, x_n$) from the dataset with shared attributes/objects. This seems like a very strong, and potentially expensive, assumption during training. The confusion about its use at inference (see Q2/Q3) makes this component's contribution difficult to assess.

4. The experimental results are not sufficiently convincing. The most recent comparison method included is from 2024 and only one such method is considered. The authors have missed many of the latest approaches in the CZSL field, which should be included for a more comprehensive and fair comparison.

[a] Unified Framework for Open-World Compositional Zero-shot Learning, WACV 2025.

[b] MRSP: Learn Multi-representations of Single Primitive for Compositional Zero-Shot Learning, ECCV 2024.

[c] Visual primitives as words: Alignment and interaction for compositional zero-shot learning, PR 2025.

[d] Refiner: Fine-grained Cross-modal Concepts Refinement for Compositional Zero-Shot Learning, ICASSP 2025.

[e] Learning Clustering-based Prototypes for Compositional Zero-Shot Learning, ICLR 2025.

**Questions:**

None

---

### Note · Authors · 2025-12-23

I have read and agree with the venue's withdrawal policy on behalf of myself and my co-authors.